

# Gated attention based generative adversarial networks for imbalanced credit card fraud detection

Jiangmeng Ge[1], Lanxiang Yin[2], Shiqing Zhang[1] and Xiaoming Zhao[1]

[1] Institute of Intelligent Information Processing, Taizhou University, Taizhou, Zhejiang, China
[2] School of Business, Taizhou University, Taizhou, Zhejiang, China

## ABSTRACT

Credit card fraud detection is highly important to maintain financial security. However, it is challenging to train suitable models due to the class imbalance in credit card transaction data. To address this issue, this work proposes a novel deep learning framework, gated attention-based generative adversarial networks (GA-GAN) for credit card fraud detection in class-imbalanced data. GA-GAN integrates GAN and the gated attention mechanism to generate high-quality synthetic data that realistically simulates fraudulent behaviors. Experimental results on two public credit card datasets demonstrate that GA-GAN outperforms state-of-the-art methods on credit card fraud detection tasks in class-imbalanced data, indicating the advantage of GA-GAN. The code is publicly available at https://github.com/Gejiangmeng/gagan/tree/main.

## INTRODUCTION

The advent of online shopping and mobile payments has led to a significant rise in the use of credit cards as a primary payment method. This shift has concurrently resulted in a notable increase in both the severity and complexity of credit card fraud. In this sense, how to accurately detect credit card fraud is highly important to maintain financial security. Traditional rule-based fraud detection methods (*Ming et al., 2024*) are insufficient to handle the increasingly complex fraudulent activities. These traditional methods struggle to fully comprehend and replicate the intricate and diverse patterns of fraud, making it challenging to protect consumer interests and maintain economic stability (*Shi & Zhao, 2023*). Therefore, it is necessary to develop advanced approaches to accurately detect credit card fraud.

Recently, machine learning methods have shown significant advantages over traditional rule-based methods (*Zhao & Guan, 2023*) for credit card fraud detection, since they can automatically extract useful features from large amounts of high-dimensional data (*Ding et al., 2024*). Representative machine learning methods are random forests (*Xuan et al., 2018*), support vector machines (SVM), long short-term memory (LSTM) (*Jurgovsky et al., 2018*), distributed neural network (DDNN) (*Lei et al., 2023*), and so on. However, due

Corresponding authors
Lanxiang Yin, zmjxylx@tzc.edu.cn
Shiqing Zhang, tzczsq@163.com

to the class imbalance issue in credit card fraud data, these methods often exhibit limited performance. Specifically, the volume of normal transaction data far outweighs that of fraudulent transactions.

To alleviate the above class imbalance issue, generating synthetic data that closely resemble real-world data, may represent a possible solution. To this end, representative methods include traditional statistical techniques such as Bayesian networks (*Young, Graham & Penny, 2009*) and recently-developed deep learning techniques such as generative adversarial networks (GAN) (*Goodfellow et al., 2014*), autoencoder (AE) (*Zhou & Paffenroth, 2017*), variational autoencoder (VAE) (*Pinheiro Cinelli et al., 2021*). Among them, GAN has become an active topic for generating synthetic data in the field of credit card fraud detection. However, these data generation methods still have a limitation. Specifically, they have difficulty in generating high-quality synthetic data that realistically simulate fraudulent behaviors when confronted with intricate fraud patterns. Consequently, how to create high-quality synthetic data for credit card fraud detection remains a significant challenge.

To address the above challenge, this work proposes a novel deep learning framework integrating gated attention with generative adversarial networks (GA-GAN) for credit card fraud detection in class imbalanced data. The proposed GA-GAN, consisting of four primary components: training the generator, training the discriminator, iterative training and the gated attention unit, aims to generate high-quality synthetic data for enlarging the original fraudulent transactions data, and promoting the performance of credit card fraud detection. Extensive experiments are implemented on two public datasets, and the results demonstrate the effectiveness of the proposed GA-GAN on credit card fraud detection tasks in class imbalanced data.

To sum up, the main contributions of this work are three-fold:

(1) A novel deep learning framework called GA-GAN equipped with a gated attention mechanism is proposed for credit card fraud detection in class imbalanced data.

(2) GA-GAN is capable of generating high-quality synthetic data that realistically simulate fraudulent behaviors, resulting in enlarging the original fraudulent transactions data and promoting the performance of credit card fraud detection.

(3) Extensive experiments on two public datasets demonstrate the validity of the proposed GA-GAN on credit card fraud detection tasks in class imbalanced data.

## RELATED WORK

### Credit card fraud detection

Early works often employ machine learning models such as anomaly detection algorithms to identify fraudulent transactions like credit card fraud. To this end, the popular traditional machine learning methods are random forest (RF), support vector machine (SVM), hidden Markov model (HMM) (*Iyer et al., 2011*), logistic regression (LR) (*Kulkarni & Ade, 2016*), and so on. Recently, due to the excellent ability of feature learning, deep learning methods have been leveraged to automatically learn relevant features from raw financial fraud data for credit card fraud detection. The typical deep learning architectures, such

as convolutional neural networks (CNN) (*Fu et al., 2016*), recurrent neural networks (RNN) (*Lin et al., 2021*), long short-term memory (LSTM) (*Jurgovsky et al., 2018*), and gated recurrent unit (GRU) (*Xie et al., 2024*), have been successfully applied to credit card fraud detection.

## GAN-based generation of synthetic data

GAN (*Goodfellow et al., 2014*) is a conventional unsupervised deep learning framework. The key idea of GAN is the adversarial training process, in which the generator and the discriminator are trained simultaneously in a competitive setting. Recently, GAN and its variants have been widely utilized to generate synthetic samples that are highly realistic in various domains. *Zhao et al. (2021)* presented a conditional table GAN architecture (CTAB-GAN) to generate diverse data types including a mix of continuous and categorical variables. *Jeon et al. (2022)* developed a general time series synthesis based on GAN capable of synthesizing regular and irregular time series data. *Zhao et al. (2024b)* designed a simultaneous generation and anomaly detection with GAN (SGAD-GAN). *Zhang et al. (2023)* proposed a CCFD-GAN equipped with a complementary distribution-based penalty mechanism for credit card fraud detection.

# OUR METHOD

## Problem definition

For credit card fraud detection in class imbalanced data, given a transaction dataset $P_{data}(X)$ containing all transaction data, $P_{data}(X)$ represents a sequence of samples $X_{1:n}$ for $n$ transactions. Each sample $X_i = \{x_i^1, x_i^2, \ldots, x_i^m\}$ has a feature dimension of $m$. Our task is to detect a sample $X_i$ is whether fraudulent or not.

## Model architecture

This work proposes a novel GAN framework called GA-GAN equipped with a gated attention mechanism. The proposed GA-GAN framework comprises of three critical steps: data preprocessing, GA-GAN for data generation, and fraud detection with fully-connected (FC) networks. The overall framework of the proposed GA-GAN for credit card fraud detection in class imbalanced data is shown in Fig. 1.

### Data preprocessing

Data preprocessing mainly involves a data normalization, as defined as:

$$x' = \frac{x - min(x)}{max(x) - min(x)},$$ (1)

where $x$ represents the original data, $x'$ represents the normalized data, $min(x)$ and $max(x)$ are the minimum and maximum values of $x$, respectively.

### GA-GAN for data generation

When using GA-GAN for data generation, we adopt training data to train our GA-GAN for producing synthetic data. To this end, four primary components: training the generator,

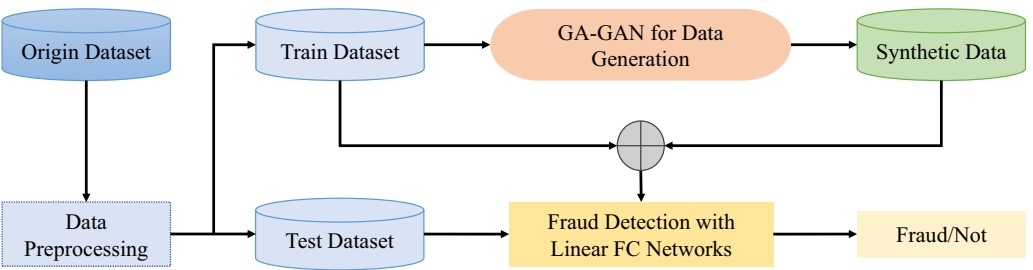

**Figure 1  The overall framework of the proposed GA-GAN for credit card fraud detection.**

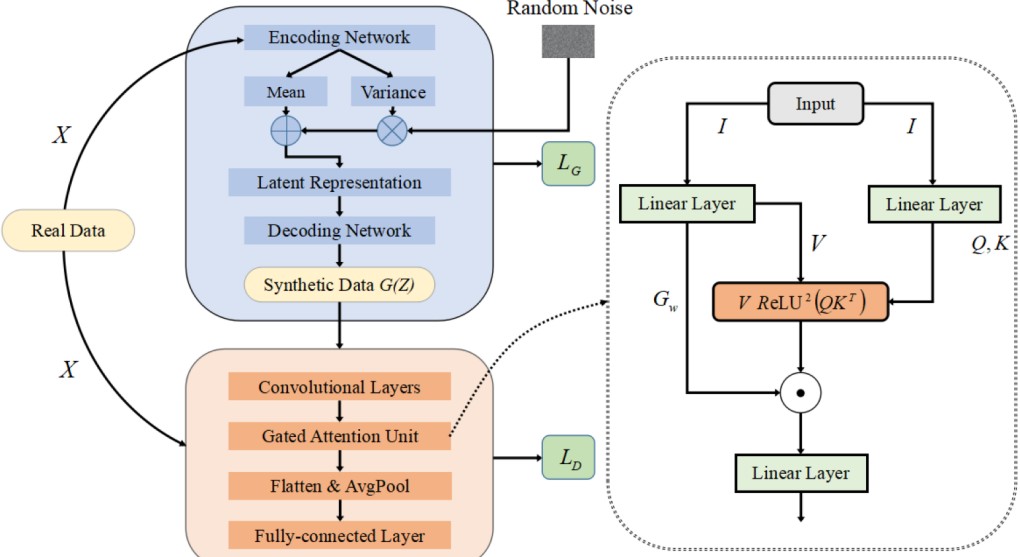

**Figure 2  The structure of our GA-GAN for data generation.**

training the discriminator, iterative training and gated attention unit, are included, as depicted in Fig. 2. These four components are described below.

(1) Training the generator

Motivated by the advantage of variational autoencoder (VAE) (*Pinheiro Cinelli et al., 2021*), we design a VAE-based generator $G$ with a similar structure to VAE. Specifically, the training data $X$ is initially fed into a encoding network which maps $X$ into a latent representation space. The encoding network consists of two FC layers, in which the first layer is equipped with a ReLU activation function to produce sparse activations, and the second layer provides the mean value $\mu$ and the variance value $\sigma^2$, respectively.

Then, based on the $\mu$ and $\sigma^2$, a random noise $Z = \{z^1, z^2, \ldots, z^m\}$ with a feature dimension of $m$ is reparameterized to generate latent representations. Finally, a decoding network, similar to the encoding network, is leveraged to produce synthetic credit card transactions data that can deceive the discriminator. To ensure that the learned distribution accurately approximates the prior distribution of input data, the generator aims to minimize

the following loss function:

$$L_G = min(-E_{X \sim P_{data}(X)}[\log p_\theta(X|G(Z))] + \beta \cdot D_{KL}(q_\phi(G(Z))|X \| p(G(Z)))), \qquad (2)$$

where $p_\theta(X|G(Z))$ represents the likelihood of the data $X$ for the obtained output $G(Z)$ of the generator $G.D_{KL}(q_\phi(G(Z)|X)\|p(G(Z)))$ is the Kullback–Leibler (KL) divergence between the approximate posterior $q_\phi(G(Z)|X)$ and the prior distribution $p(G(Z))$. The parameter $\beta$ controls a trade-off between the reconstruction fidelity and the regularization term.

(2) Training the discriminator

The real data $X$ and synthetic data $G(Z)$ are mixed together and fed into the model's discriminator. In the discriminator, input data are initially fed into three convolutional layers for feature extraction, each of which has a convolutional kernel of $1 \times 3$. Then, a gated attention unit is designed for feature enhancement, which focuses on capturing key clues while suppressing redundant details in feature representations. Subsequently, flattening and average-pooling (AvgPool) operations are performed, followed by three FC layers. The loss function of the discriminator $D$ is defined as:

$$L_D = min(-E_{X \sim P_{data}(X)}[logD(X)] - E_{Z \sim P_Z(Z)}[log(1 - D(G(Z)))]). \qquad (3)$$

(3) Iterative training

The aforementioned two steps are repeated iteratively, continuously updating the parameters of both the generator $G$ and the discriminator $D$, until the model converges. The convergence is reached when the generator $G$ can produce synthetic fraudulent transactions that sufficiently deceive the discriminator $D$, while $D$ is capable of accurately identifying the real fraudulent transactions and the synthetic ones. In this sense, the purpose of the iterative training process is to solve the resulting minimax optimization problem:

$$\max_G \min_D L(D, G) = E_{X \sim P_{data}(X)}[logD(X)] + E_{Z \sim P_Z(Z)}[log(1 - D(G(Z)))]. \qquad (4)$$

(4) Gated attention unit

Inspired by the advantage of gated attention mechanisms (*Xue, Li & Zhang, 2020*; *Niu, Zhong & Yu, 2021*; *Hua et al., 2022*), we design a gated attention unit (GAU) based on gated attention mechanisms, as illustrated in Fig. 2. The detailed steps of GAU is listed below.

Initially, a linear layer is utilized to transform an input data $I$ to a new tensor $Z_h$ with a feature dimension of $2H$, where $H$ is the number of hidden layer neurons. This process is expressed as:

$$Z_h = \text{Linear}(I). \qquad (5)$$

Then, through a chunk operation, $Z_h$ is split into two tensors: $V$ and $G_w$, each of which has the same feature dimension $H$. Among them, $V$ denotes the value matrix, and $G_w$ is the gated weights. Additionally, the input data $I$ is fed into a linear layer to produce the query $(Q)$ and key $(K)$ matrices with a dimension of $H$, as defined as:

$$(Q, K) = \gamma \cdot \text{Linear}(I) + \beta, \qquad (6)$$

where $\gamma$ and $\beta$ are are learnable scaling factors and biases, respectively.

Next, we compute the similarity between each query $Q$ and key $K$, and then obtain the attention scores $A$ through a ReLU activation function and a squared operation, as expressed as:

$$A = ReLU^2 \left( \frac{Q \cdot K^T}{\sqrt{H}} \right). \tag{7}$$

Subsequently, a dropout layer is employed to prevent overfitting. In this case, the attention matrix $A$ aims to capture the relationships between the elements of the sequence in a refined manner. Then, multiplying each value vector by its corresponding attention weight is performed to adjust the contribution of the value matrix $V$, as expressed as:

$$\text{Att\_scores} = \text{Dropout}(A) \cdot V, \tag{8}$$

where the weighted output $Att\_scores$ reflects the relationships between different elements in input data. Then, $Att\_scores$ is multiplied in an element-wise way by the corresponding gating vector $G_w$ for producing the final weighted values, as defined as:

$$O_w = \text{Att\_scores} \odot G_w. \tag{9}$$

Finally, a linear layer is leveraged to transform the feature dimension of $O_w$ into the appropriate dimensionality for the next layer.

### Output of fraud detection

Based on the mixed input data including real data and synthetic data, a linear FC network consisting of four FC layers is used to train a fraud detection model, and identify a given transaction sample from testing dataset is whether fraud or not. To train the fraud detection model, a binary cross-entropy loss function $L_{FC}$ is utilized, as defined as:

$$L_{FC} = -\frac{1}{N+U} \sum_{i=1}^{N+U} \left[ Y_i \log f(X_i) + (1-Y_i)\log(1-f(X_i)) \right], \tag{10}$$

where $N$ is the number of original data, $U$ is the number of synthetic data, and $Y_i$ represents the true label for each sample. $f(X_i)$ denotes the predicted probability of the $i-th$ transaction data being fraudulent.

## EXPERIMENT STUDY

### Datasets and evaluation metric

European dataset (*Machine Learning Group - ULB, 2017*): This dataset is available at https://www.kaggle.com/datasets/mlg-ulb/creditcardfraud. It contains 284,807 transaction records of European credit card holders in September 2013, covering transactions within two days. In this dataset, only 0.172% are fraudulent (492 transactions in total). These data are mainly Principal Component Analysis(PCA)-transformed numeric variables. Due to confidentiality issues, 28-dimensional features are principal components obtained by PCA, whereas the original features such as 'Time' and 'Amount' are provided.

Card Fraud dataset (*R, 2021*): This dataset is available at https://www.kaggle.com/datasets/dhanushnarayananr/credit-card-fraud. It collects 1,000,000 transactions. Each transaction

consists of seven features, indicating whether a transaction is fraudulent. Specially, it includes the distance from the location of the bank card transaction to the cardholder's home, the geographical distance between the current transaction and the last transaction, and five other features. Note that this dataset is usually used for competition.

Following in *Zhao et al. (2024a)* and *Fanai & Abbasimehr (2023)*, four typical evaluation metrics such as Precision, Recall, F1-Score,and Area Under the Curve (AUC) are leveraged to evaluate the performance of all used methods on credit card fraud detection tasks in class imbalanced data.

## Implementation details

All used methods are performed in a Pytorch 3.10 framework and a single NVIDIA 4090 GPU with 24 GB of RAM. The Adam optimizer is adopted with a learning rate of 0.002. The discriminator uses the LeakyReLU activation function with a slope of 0.2. In the GAU, the number of hidden layer neurons is 128. The batch size is 1024, and the epoch number is 500. The binary cross-entropy loss function is used for credit card fraud detection in class imbalanced data.

Each used dataset is divided into two parts: 80% for training, and 20% for testing. On the European dataset, fraud samples in the divided training set are enlarged by GA-GAN from 383 samples to 113,731 samples. Likewise, on the Card Fraud dataset, fraud samples in the divided training set are enlarged by GA-GAN from 69,922 samples to 336,501 samples.

## Baseline

To demonstrate the effectiveness of our GA-GAN, we compare our method with the following five baseline models, as listed below:

(1) Logistic regression (*Kulkarni & Ade, 2016*): LR adopts the logistic function to predict the probability that an instance belongs to one of two classes.

(2) Support vector machines (*Lu & Ju, 2011*): SVM aims to find the optimal separating hyperplane that maximizes the margin between different classes. The typical RBF kernel is used for SVM.

(3) Random forests (*Xuan et al., 2018*): RF aims to combine the predictions of multiple decision trees to improve the predictive accuracy.

(4) Long short-term memory network (LSTM) (*Jurgovsky et al., 2018*): LSTM is a deep learning method capturing long-term dependencies of sequence data.

(5) Generative adversarial networks (*Goodfellow et al., 2014*): GAN is a deep learning approach to generative modeling by pitting two neural networks against each other in a competitive setting.

## Experimental results and analysis

Table 1 presents a performance comparison of all used methods such as LR, SVM, RF, LSTM, GAN, and GA-GAN on these two public datasets. The results in Table 1 show that the proposed GA-GAN model obtains better performance than other methods. In particular, GA-GAN provides the highest precision (0.903), recall (0.848), F1-Score (0.875) and AUC (0.929) on the European dataset. Similarly, on the credit fraud card dataset GA-GAN yields the best precision (0.987), recall (0.978), F1-Score (0.982) and AUC (0.972). This shows the

**Table 1** Performance comparison of different methods on the European and Credit Fraud datasets.

| Database | Metric | LR | SVM | LSTM | RF | GAN | GA-GAN |
|---|---|---|---|---|---|---|---|
| | Precision ↑ | 0.646 | 0.733 | 0.774 | 0.772 | 0.830 | **0.903** |
| European | Recall ↑ | 0.697* | 0.728* | 0.750* | 0.745* | 0.819** | **0.848** |
| Dataset | F1-Score ↑ | 0.671* | 0.730* | 0.762** | 0.758** | 0.824*** | **0.875** |
| | AUC ↑ | 0.848* | 0.741* | 0.766* | 0.872* | 0.814* | **0.929** |
| | Precision ↑ | 0.893 | 0.847 | 0.984 | 0.985 | 0.972 | **0.990** |
| Card Fraud | Recall ↑ | 0.605* | 0.746* | 0.801* | 0.772** | 0.852** | **0.988** |
| Dataset | F1-Score ↑ | 0.721* | 0.793** | 0.883* | 0.870** | 0.908*** | **0.989** |
| | AUC ↑ | 0.967* | 0.956* | 0.970* | 0.962* | 0.950* | **0.983** |

**Notes.**
† Significant level:
* $P < 0.05$.
** $P < 0.01$.
*** $P < 0.001$.
Bold values denote the best performance.

advantage of our GA-GAN over these baseline methods. The main reason is that GA-GAN is effective for data augmentation on credit fraud card detection tasks. Moreover, the order of other baseline methods is GAN, RF, LSTM, SVM and LR. This shows GAN-based methods outperform other conventional methods without data augmentation. This may be attributed to the effectiveness and potential of GAN-based methods for generating synthetic data on credit fraud card detection tasks in class imbalanced data.

To further evaluate the robustness of the obtained results, we perform a statistical significance testing on all used models. Specifically, the paired t-tests are conducted to compare the performance metrics of GA-GAN with each of the baseline methods across both datasets. The obtained $P$-values from these t-tests indicate that the improvements achieved by GA-GAN are statistically significant at different levels. The smaller the obtained $P$-values, the more significant the achieved results. As shown in Table 1, compared with other methods, there is a large statistical significance in F1-Score for GA-GAN since its $P$-value is less than 0.001. Likewise, since the obtained $P$-value of GA-GAN is less than 0.01, a relatively stronger statistical significance in recall is shown for GA-GAN in comparison with other methods. Similarly, a relatively lower statistical significance in AUC is shown for GA-GAN since its obtained $P$-value is less than 0.05. However, the paired $T$-test yields a significance level with a $P$-value more than 0.05 for precision, indicating that no significant difference in precision is observed. This may be attributed to the imbalanced data distribution.

## Compared with other works

To further show the advantage of the proposed GA-GAN, we compare our method with several recently-reported works on the European dataset, which are similar to our experiment settings, as listed below. It is noted that we fail to find existing relevant literatures for comparison on the Credit Fraud dataset since this dataset is usually used for competition.

**Table 2  Performance comparison between ours and other works on European dataset.**

| Method | F1-Score ↑ | Recall ↑ | Precision ↑ | AUC ↑ |
|---|---|---|---|---|
| SMEM(2021) | 0.833 | 0.777 | 0.889 | 0.813 |
| AE-DNN(2023) | 0.857 | 0.839 | 0.857 | 0.907 |
| RUS-CAE(2023) | 0.802 | 0.752 | 0.860 | 0.902 |
| SAGAN(2024) | 0.847 | 0.840 | 0.855 | 0.857 |
| Ours | **0.875** | **0.848** | **0.903** | **0.929** |

**Notes.**
  Bold values denote the best performance.

SMEM (*Forough & Momtazi, 2021*): a sequential modeling-based ensemble model that combines LSTMs and GRUs for detecting fraudulent transactions.

AE-DNN (*Fanai & Abbasimehr, 2023*): a two-stage deep neural network that integrates a deep AE with deep classifiers for credit card fraud detection.

RUS-CAE (*Salekshahrezaee, Leevy & Khoshgoftaar, 2023*): integrating a data sampling strategy called Random Undersampling (RUS) with a feature extraction scheme based on a Convolutional Autoencoder (CAE), followed by an ensemble classifier for credit card fraud detection.

SAGAN (*Zhao et al., 2024a*): a deep learning model that integrates self-attention mechanisms with GAN for credit card fraud detection.

Table 2 provides a performance comparison between our method and other works on the European dataset. The results in Table 2 indicate that our method achieves better performance on credit card fraud detection tasks in class imbalanced data than SMEM, AE-DNN, SAGAN and RUS-CAE in terms of F1-score, recall and AUC. This shows the superiority of our method to other methods.

## Ablation study
### Effect of GAU

To demonstrate the effectiveness of the used GAU in the discriminator, Fig. 3 shows a comparison of the obtained results on the European and Card Fraud datasets, when using GAU or not. When removing the GAU, our method is implemented by a FC layer.

As shown in Fig. 3, the obtained results with GAU clearly outperform the results achieved without GAU. In particular, on the European dataset, the obtained F1-score without GAU is 0.754. In contrast, the obtained F1-score with GAU is 0.854, indicating a increase of 0.1 over the case without GAU. On the Card Fraud dataset, the achieved F1-score with the GAU is 0.989, yielding an improvement of 0.37 over the case without GAU. This indicates that the GAU enhances the model's ability to distinguish between fraudulent and non-fraudulent transactions.

### Effect of the structure of generator

This work designs a VAE-based generator with the similar structure to VAE (*Pinheiro Cinelli et al., 2021*). To validate the effectiveness of VAE-generator in our GA-GAN, Fig. 4 illustrates a performance comparison of detection results on two datasets. It can be seen from Fig. 4 that VAE-based generator obtains a significant improvement in terms of

**PeerJ** Computer Science ___________

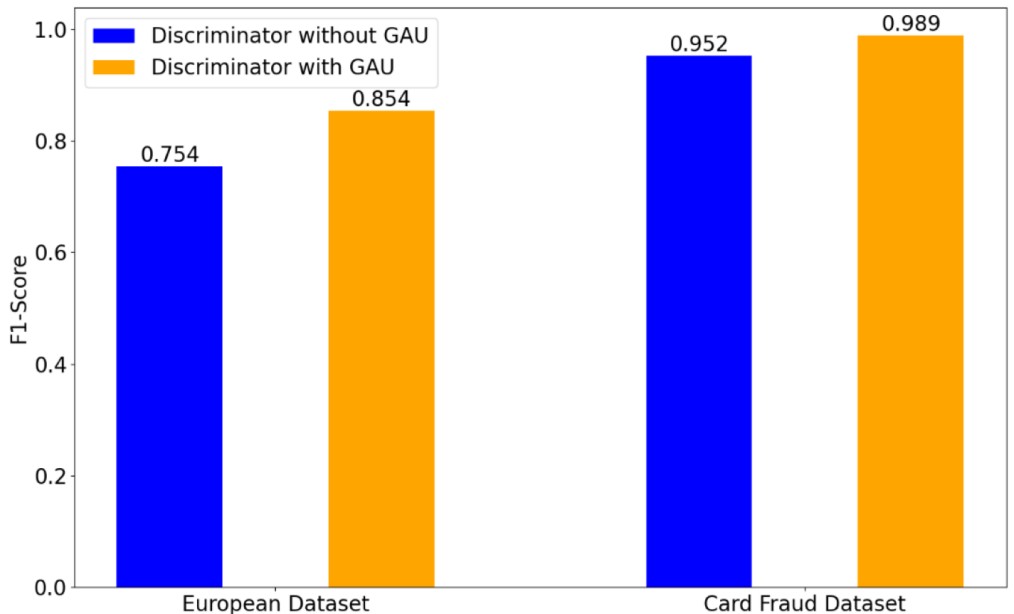

**Figure 3** Effect of GAU in the discriminator on credit card fraud detection tasks.

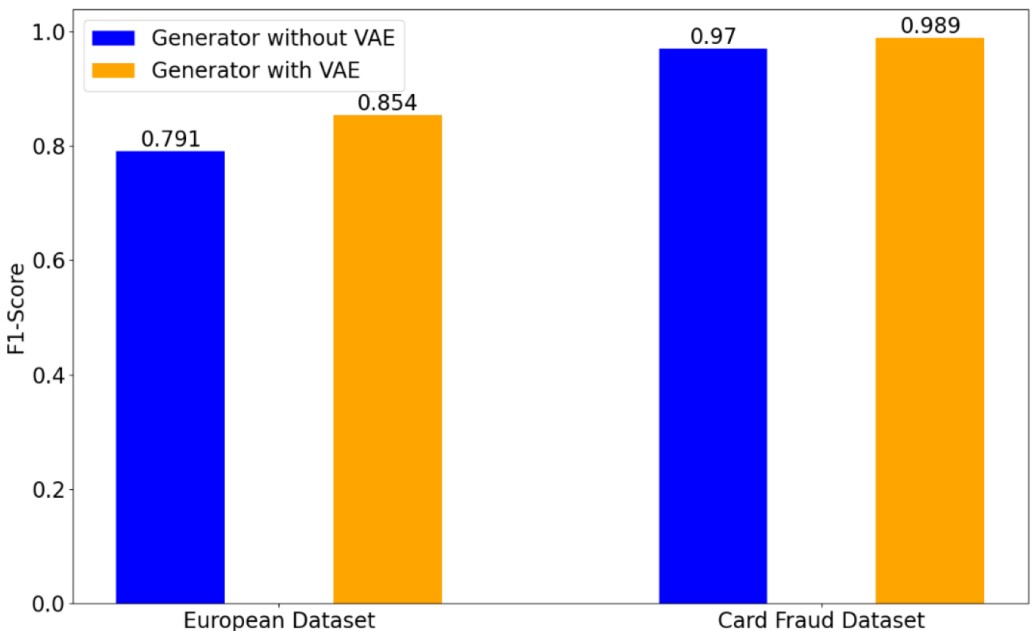

**Figure 4** Effect of the structure of generator on credit card fraud detection tasks.

F1-score over the original generator (*Goodfellow et al., 2014*) without VAE, in which the generator is implemented by three FC layers. In particular, on the European dataset, VAE-based generator gives a F1-score of 0.854, whereas the generator without VAE yields

a F1-score of 0.791. On Card Fraud dataset, VAE-based generator improves a F1-score from 0.970 to 0.989 compared with the generator without VAE. This demonstrates the advantage of VAE-based generator in our GA-GAN for data generation.

### Effect of synthetic data

To intuitively demonstrate the impact of synthetic data generated by GA-GAN on credit card fraud detection tasks, we compare the confusion matrices obtained from training with and without synthetic data across two datasets, as shown in Fig. 5. The results indicate that on the original European dataset (Fig. 5A), our model achieves a F1-score of 0.770, suggesting a certain level of false negatives (missed fraudulent transactions), where actual fraud cases are not effectively identified. In this case, 46 fraud cases are not correctly classified. However, after training with synthetic data generated by GA-GAN (Fig. 5B), the obtained F1-score is improved significantly to 0.875. In this case, 24 fraud cases are not correctly classified. This demonstrates that GA-GAN not only enhances the identification of fraudulent transactions but also maintains high accuracy for normal transactions. Similarly, in the Credit Fraud dataset, the original F1-score is 0.983 (Fig. 5C), which is promoted to 0.989 (Fig. 5D) when using the augmented dataset with GA-GAN, approaching near-perfect classification. In this case, the number of fraud cases, which are not correctly recognized, is reduced from 627 to 382. This further validates the effectiveness of GA-GAN in reducing both false positives and false negatives. Moreover, these findings highlight the potential of GA-GAN as a powerful tool in addressing class-imbalanced credit card fraud detection problems. By generating high-quality synthetic data, GA-GAN effectively mitigates the challenges posed by data imbalance, significantly enhancing the overall performance of our model.

## CONCLUSION AND FUTURE WORK

In this work, we propose a novel GAN framework called GA-GAN equipped with a gated attention mechanism for credit card fraud detection in class imbalanced data. The proposed GA-GAN contains three key steps: data preprocessing, GA-GAN for data generation, and fraud detection with FC networks. GA-GAN is capable of generating high-quality synthetic data that realistically simulate fraudulent behaviors. Experimental results on two public credit card fraud datasets demonstrate the advantage of the proposed GA-GAN. It is pointed out that GA-GAN has a relatively high computational complexity with a network parameter of approximately 2M, and a FLOP(Floating Point Operations Per Second) of 9.06G. In this sense, it is interesting to further reduce the computational complexity of GA-GAN in future when applying to large-scale credit card transaction data in real-world scenarios. Besides, it is noted that the raw tabular data related to credit card fraud detection may not inherently contain semantic information, resulting in the difficulty of investigating the interpretability of our model in the context of tabular data, as shown in *Borisov et al. (2024)*. Nevertheless, it is meaningful to explore the interpretability of our model in the context of tabular data in future.

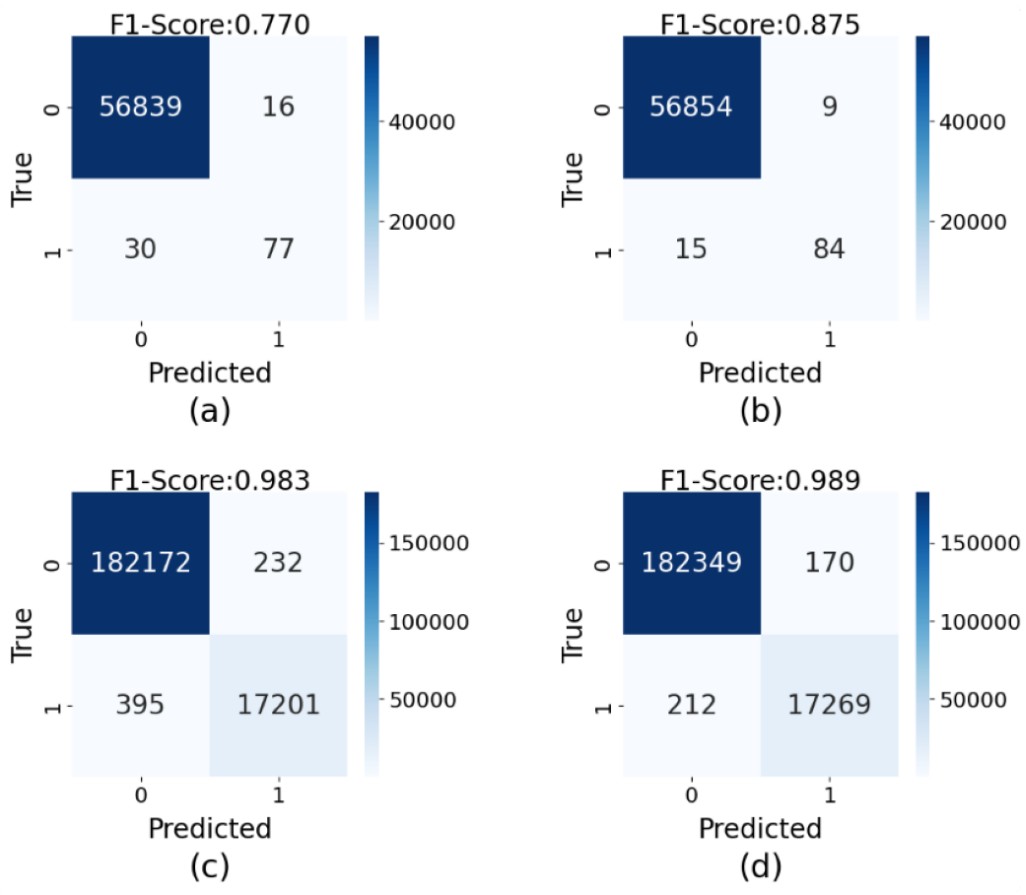

**Figure 5** The confusion matrices of detection results on the testing data of two datasets: (A) the results on the original European dataset, (B) the results on the enlarged European dataset with GA-GAN, (C) the results on the original Card Fraud dataset, (D) the results on the enlarged Card Fraud dataset with GA-GAN.

### Funding

This work was supported by National Natural Science Foundation of China (NSFC) under Grant No. 62276180. The funders had no role in study design, data collection and analysis, decision to publish, or preparation of the manuscript.

### Grant Disclosures

The following grant information was disclosed by the authors:
National Natural Science Foundation of China (NSFC): No. 62276180.

### Competing Interests

The authors declare there are no competing interests.

## Author Contributions

- Jiangmeng Ge conceived and designed the experiments, performed the experiments, performed the computation work, prepared figures and/or tables, authored or reviewed drafts of the article, and approved the final draft.
- Lanxiang Yin analyzed the data, authored or reviewed drafts of the article, project administration, and approved the final draft.
- Shiqing Zhang conceived and designed the experiments, prepared figures and/or tables, authored or reviewed drafts of the article, and approved the final draft.
- Xiaoming Zhao performed the computation work, authored or reviewed drafts of the article, funding acquisition, and approved the final draft.

## Data Availability

The code and data are available at Zenodo: Gejiangmeng. (2025). Code and Datasets for GAGAN. Zenodo. Available at https://doi.org/10.5281/zenodo.15265998.

## Supplemental Information

Supplemental information for this article can be found online at http://dx.doi.org/10.7717/peerj-cs.2972#supplemental-information.

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
