# Peer review of "Gated attention based generative adversarial networks for imbalanced credit card fraud detection"

_PeerJ Computer Science, doi:10.7717/peerj-cs.2972_

## Round 0.1 · original submission · Major Revisions

Please address all of the reviewer comments.

**Language Note:** PeerJ staff have identified that the English language needs to be improved. When you prepare your next revision, please either (i) have a colleague who is proficient in English and familiar with the subject matter review your manuscript, or (ii) contact a professional editing service to review your manuscript. PeerJ can provide language editing services - you can contact us at [email protected] for pricing (be sure to provide your manuscript number and title). – PeerJ Staff

·

Basic reporting

The manuscript is well written in clear, professional English, making it easy for readers to understand the research.
The introduction provides a good context for the study, highlighting the limitations of traditional rule-based fraud detection methods and the advantages of machine learning approaches. The literature review is comprehensive and relevant.
The structure of the manuscript conforms to the standards of PeerJ and the discipline. The paper is logically organized with clear sections, including Abstract, Introduction, Related Work, Our Method, Experiment Study, and Conclusion.
The figures are relevant, of high quality, and well labeled and described, effectively supporting the understanding of the proposed method and experimental results.
However, before final acceptance, the authors should consider the following improvements:
1.The comparison with other recent works could be expanded to include more state-of-the-art methods to further validate the superiority of GA-GAN.
2.More detailed analysis of the computational complexity and efficiency of the proposed method would be beneficial, especially in the context of large-scale credit card transaction data.
3.The authors should discuss the potential limitations of their approach and suggest possible directions for future research.

Experimental design

The authors have performed a rigorous investigation to a high technical standard. The methods are described in sufficient detail, allowing others to replicate the study.

Validity of the findings

The underlying data are robust, statistically sound, and well controlled. The conclusions are well stated, directly linked to the original research question, and appropriately limited to the supporting results.

Reviewer 2 ·

Basic reporting

1.Language and Clarity: The paper is written in clear and professional English. However, minor grammatical issues and sentence restructuring could enhance readability.

2.Introduction and Background: The authors provide a comprehensive background on fraud detection and the challenges of class imbalance. Citing state-of-the-art works like GANs, Autoencoders, and VAEs is appropriate.

3.Literature Review: The related work section is well-referenced and covers essential studies. However, adding recent studies on credit card fraud detection using deep learning would strengthen the context.

4. Figures and Tables: Figures and tables are relevant and labeled properly. Including additional visual comparisons of results using performance metrics could further support claims.

Experimental design

1.Novelty: The integration of Gated Attention with GANs for fraud detection is novel and justified. The proposed GA-GAN model design appears well-structured.

2. Data and Methods: The authors use two public datasets for evaluation, which is appropriate. The methodology is clearly explained with detailed steps on data preprocessing, model design, and parameter settings.

3. Reproducibility: The authors provide sufficient information for replication. They also mention that the code is publicly available, which enhances reproducibility.

Validity of the findings

1.Evaluation Metrics: Commonly used metrics like precision, recall, F1-score, and AUC are reported. Including additional analysis on model interpretability could add value.

2. Comparison with State-of-the-Art: The model’s performance is compared with recent baseline methods. However, further justifications on the choice of baselines and statistical significance tests would improve the validity of the results.

3. Ablation Studies: The paper includes ablation experiments to demonstrate the effectiveness of the Gated Attention module. Providing further insights into why specific modules contribute to performance improvements would be beneficial.

Additional comments

Strengths:

Novel application of Gated Attention in GANs.

Strong empirical results on credit card fraud detection datasets.

Clear and structured methodology with publicly available code.

Areas for Improvement:

Enhance the discussion on interpretability and explainability.

Provide more detailed error analysis.

Perform statistical significance testing to confirm the robustness of the results.

Recommendation: Minor Revisions

With minor adjustments, this paper has the potential for publication. Addressing the comments on interpretability, result analysis, and further clarifications will strengthen the contribution and impact of the work.

---

## Round 0.2 · accepted · Accept

Dear authors, we are pleased to verify that you have met the reviewers' valuable feedback to improve your research.

Thank you for considering PeerJ Computer Science and submitting your work.

Kind regards
PCoelho

·

Basic reporting

The authors have addressed all my concerns.

Experimental design

No

Validity of the findings

No

Additional comments

No

Reviewer 2 ·

Basic reporting

Basic reporting
1. Language and Clarity: The paper is written in clear and professional English.

2. Introduction and Background: The authors provide a comprehensive background on fraud detection and the challenges of class imbalance. Citing state-of-the-art works like GANs, Autoencoders, and VAEs is appropriate.

3. Literature Review: The related work section is well-referenced and covers essential studies.

4. Figures and Tables: Figures and tables are relevant and labeled properly.

Experimental design

Experimental design
1. Novelty: The integration of Gated Attention with GANs for fraud detection is novel and justified. The proposed GA-GAN model design appears well-structured.

2. Data and Methods: The authors use two public datasets for evaluation, which is appropriate. The methodology is clearly explained with detailed steps on data preprocessing, model design, and parameter settings.

3. Reproducibility: The authors provide sufficient information for replication. They also mention that the code is publicly available, which enhances reproducibility.

Validity of the findings

Validity of the findings
1. Evaluation Metrics: Commonly used metrics like precision, recall, F1-score, and AUC are reported.

2. Comparison with State-of-the-Art: The model’s performance is compared with recent baseline methods.

3. Ablation Studies: The paper includes ablation experiments to demonstrate the effectiveness of the Gated Attention module.

Additional comments

Strengths:

Novel application of Gated Attention in GANs.

Strong empirical results on credit card fraud detection datasets.

Clear and structured methodology with publicly available code.